# Association between Co-Morbidities and the Prevalence of Excessive Daytime Sleepiness over a Four-Year Period

**Chandima Karunanayake [1,\*], James Dosman [1], Mark Fenton [2], Donna Rennie [3], Shelley Kirychuk [1,2], Vivian Ramsden [4], Jeremy Seeseequasis [5], Sylvia Abonyi [6], Punam Pahwa [1,6] and First Nations Lung Health Project Research Team [1]**

[1] Canadian Centre for Health and Safety in Agriculture, University of Saskatchewan, 104 Clinic Place, Saskatoon, SK S7N 2Z4, Canada; james.dosman@usask.ca (J.D.); shelley.kirychuk@usask.ca (S.K.); pup165@mail.usask.ca (P.P.); firstnations.lung@usask.ca (F.N.L.H.P.R.T.)

[2] Department of Medicine, University of Saskatchewan, Royal University Hospital, 103 Hospital Drive, Saskatoon, SK S7N 0W8, Canada; mef132@mail.usask.ca

[3] College of Nursing, University of Saskatchewan, 104 Clinic Place, Saskatoon, SK S7N 2Z4, Canada; donna.rennie@usask.ca

[4] Department of Academic Family Medicine, University of Saskatchewan, West Winds Primary Health Centre, 3311 Fairlight Drive, Saskatoon, SK S7M 3Y5, Canada; viv.ramsden@usask.ca

[5] Community A, PO Box 96, Duck Lake, SK S0K 1J0, Canada; JSeeseequasis@beardysband.com

[6] Department of Community Health & Epidemiology, College of Medicine, University of Saskatchewan, 107 Wiggins Road, Saskatoon, SK S7N 5E5, Canada; sya277@mail.usask.ca

\* Correspondence: cpk646@mail.usask.ca; Tel.: +1-306-966-1647

**Abstract:** Excessive daytime sleepiness (EDS) is a common problem in general the Canadian population. It can effect day-to-day activities and is also associated with several health issues. This study aimed to investigate the association between co-morbidities and the prevalence of EDS over a four-year period in adults living in two First Nation communities. Data collected during the First Nations Lung Health Project (FNLHP) conducted in two Cree First Nation communities in Saskatchewan in 2012–2013 (Cycle 1) and 2016 (Cycle 2) were used for this analysis. There were 859 participants aged 18 years and older at baseline (Cycle 1) and 821 participants aged 18 years and older at follow-up (Cycle 2) who completed the interviewer-administered questionnaire. An Epworth Sleepiness Scale (ESS) score > 10 was considered to be abnormal and identified as a case of EDS at both time points. A multilevel logistic regression model using a generalized estimating equations approach was used to analyze the data. The prevalence of EDS at baseline (Cycle 1) was 11.2% (91/815) and 10.0% (80/803) at follow-up (Cycle 2). Based on the predicted model, longitudinal change in the prevalence of EDS was −0.11% for 358 individuals who participated in both cycles. There were 49% males at baseline and 48% males at follow-up. Multivariate regression model results revealed that crowding, shortness of breath, loud snoring, chronic lung disease, depression and gastric reflux were the main significant predictors of EDS. In addition, the interaction between sex and age was significant. Some of the co-morbid conditions were associated with EDS. Therefore, managing such conditions requires considerations in strategies to decrease the prevalence of daytime sleepiness.

**Keywords:** excessive daytime sleepiness; First Nations; snoring; chronic lung disease; depression; gastric reflux

## 1. Introduction

Excessive daytime sleepiness (EDS) is a common problem in the general population of Canada. It can affect day-to-day activities [1–3] and is often associated with serious health outcomes or co-morbidities such as diabetes [4,5], heart disease [6,7], stroke [8], depression [3,9,10], and gastroesophageal reflux disease [11]. Co-morbidity refers to the co-occurrence of two or more disorders or diseases [12]. Studies have shown that EDS is a potential risk factor for co-morbidities such as hypertension [13], diabetes [13], cardiovascular disease and mortality [14]. In addition to those, there are significant associations between chronic respiratory disease and EDS [15–18]. Motor vehicle accidents are the most dramatic consequence of EDS [19–25], with EDS accounting for 20% of total accidents. The prevalence of EDS in the general population varies from 9% to 28% [26]. Two studies of rural Canadians reported that the prevalence of subjective EDS were 15.9% [27] and 15.1% [19] in a population of adults. However, the baseline report of this study reported that the prevalence of subjective EDS was 11% in a population of First Nations adults [28].

To our knowledge, the association between co-morbidities and the prevalence of EDS in First Nations adults has not been investigated. Thus, this study aimed to investigate the association between co-morbidities and the prevalence of EDS over a four-year period among adults living in two First Nations communities.

## 2. Results

The prevalence of EDS at baseline (Cycle 1) was 11.2% (91/815) and 10.0% (80/803) at follow-up (Cycle 2). Of the 358 participants with both Epworth Sleepiness Scale (ESS) measurements, 317 who did not have EDS at baseline and 41 did have EDS at baseline. There were 24 subjects with EDS at follow-up (Cycle 2) who had no EDS at baseline survey (Cycle 1) [29]. The incidence was 24/317 = 7.6% [29]. The average age of participants during Cycle 1 was 35.2 years, with a standard deviation of 14.1 years, ranging from 18 to 81 years, with 49% of males at baseline (Cycle 1). The average age of participants during Cycle 2 was 38.3 years, with a standard deviation of 14.9 years, ranging from 18 to 89 years, with 48% of males at follow-up (Cycle 2). Baseline demographics and other covariates by ESS [30–33] score (normal vs abnormal) are presented in Table 1. The variables age, shortness of breath, loud snoring, bronchitis attack, chronic lung diseases, sinus problem, heart problem, tuberculosis, depression and gastric reflux were associated with EDS at baseline (Cycle 1). Multivariate model results are reported in Table 2. We observed that crowded living conditions, shortness of breath, loud snoring, chronic lung disease, depression and gastric reflux were significant predictors of EDS. In addition, the interaction between sex and age was significant. Quasi likelihood under the independence model criterion (QIC) without an interaction term was 695.4522, and QIC with an interaction term was 690.4543.

**Table 1.** Baseline (Cycle 1) demographics and other information by Excessive Daytime Sleepiness (EDS) (*n* = 815) for 18 years and older participants.

| Variable | Excessive Daytime Sleepiness | | *p*-Value ** |
|---|---|---|---|
| | EDS (Yes) n (%) | EDS (No) n (%) | |
| **Sex** | | | |
| Male | 40 (44.0) | 363 (50.1) | 0.2815 |
| Female | 51 (56.0) | 361 (49.9) | |
| **Age groups, in years** | | | |
| <35 years | 45 (49.4) | 432 (59.7) | 0.0017 |
| 35–55 years | 25 (27.5) | 232 (32.0) | |
| ≥56 years | 21 (23.1) | 60 (8.3) | |
| **Education level** | | | |
| Less than high school | 50 (55.5) | 342 (47.3) | 0.1446 |
| Completed high school | 15 (16.7) | 190 (26.3) | |
| Higher than high school | 25 (27.8) | 191 (26.4) | |

**Table 1.** *Cont.*

| Variable | Excessive Daytime Sleepiness | | *p*-Value ** |
|---|---|---|---|
| | EDS (Yes) n (%) | EDS (No) n (%) | |
| Marital Status | | | |
| Married/Common Law | 34 (37.8) | 285 (40.5) | 0.6212 |
| Single/separated/widowed/divorced | 56 (62.2) | 419 (59.5) | |
| Body Mass Index (BMI) | | | |
| Normal: BMI < 25 | 28 (31.1) | 239 (33.8) | 0.8842 |
| Overweight: BMI 25–29.99 | 28 (31.1) | 212 (29.9) | |
| Obese: BMI ≥ 30 | 34 (37.8) | 257 (36.3) | |
| Smoking status | | | |
| Current smoker | 64 (70.3) | 575 (79.4) | 0.1978 |
| Ex-smoker | 16 (17.6) | 82 (11.3) | |
| Never smoker | 11 (12.1) | 67 (9.3) | |
| Alcohol use (5 or more drinks on one occasion) | | | |
| Never | 21 (23.9) | 154 (21.4) | 0.3344 |
| Occasionally | 36 (40.9) | 250 (34.7) | |
| Regularly | 31 (35.2) | 317 (43.9) | |
| Employment | | | |
| Some work | 21 (23.1) | 206 (28.6) | 0.2507 |
| Unemployed | 70 (76.9) | 515 (71.4) | |
| Crowding | | | |
| >1 person per room | 34 (39.5) | 211 (30.7) | 0.1347 |
| ≤1 person per room | 52 (60.5) | 476 (69.3) | |
| Money left over at end of the month | | | |
| Some money | 37 (43.5) | 336 (49.0) | 0.3174 |
| Just enough money | 18 (21.2) | 165 (24.1) | |
| Not enough money | 30 (35.3) | 184 (26.9) | |
| Co-morbidities | | | |
| Shortness of breath (SOB) | | | |
| Yes | 69 (75.8) | 359 (49.6) | <0.0001 |
| No | 22 (24.2) | 365 (50.4) | |
| Loud snoring | | | |
| Yes | 21 (23.1) | 110 (15.2) | 0.0783 |
| No | 70 (76.9) | 612 (84.8) | |
| Bronchitis attack | | | |
| Yes | 37 (46.2) | 178 (28.0) | 0.0027 |
| No | 43 (53.8) | 457 (72.0) | |
| Chronic lung diseases * | | | |
| Yes | 40 (44.0) | 184 (25.4) | 0.0016 |
| No | 51 (56.0) | 540 (74.6) | |
| Sinus problem | | | |
| Yes | 38 (47.5) | 188 (29.0) | 0.0018 |
| No | 42 (52.5) | 460 (71.0) | |
| Heart Problem | | | |
| Yes | 17 (20.2) | 61 (8.8) | 0.0096 |
| No | 67 (79.8) | 634 (91.2) | |
| Tuberculosis | | | |
| Yes | 12 (16.0) | 43 (6.8) | 0.0388 |
| No | 63 (84.0) | 586 (93.2) | |
| Asthma | | | |

**Table 1.** *Cont.*

| Variable | Excessive Daytime Sleepiness | | *p*-Value ** |
|---|---|---|---|
| | EDS (Yes) n (%) | EDS (No) n (%) | |
| Yes | 17 (18.7) | 118 (16.3) | 0.5707 |
| No | 74 (81.3) | 606 (83.7) | |
| **Diabetes** | | | |
| Yes | 16 (18.6) | 81 (11.5) | 0.0954 |
| No | 70 (81.4) | 620 (88.5) | |
| **Depression** | | | |
| Yes | 28 (34.6) | 121 (17.3) | 0.0016 |
| No | 53 (65.4) | 577 (82.7) | |
| **Gastric Reflux** | | | |
| Yes | 25 (27.8) | 111 (15.5) | 0.0081 |
| No | 65 (72.2) | 603 (84.5) | |

\* We used the term "chronic lung diseases" to include one or more of emphysema, chronic bronchitis, chronic cough, chronic phlegm and chronic obstructive pulmonary disease (COPD). \*\* *p*-value from Rao–Scott chi-square tests for significant difference in proportion with Epworth Sleepiness Scale (ESS) score between levels of each variable.

**Table 2.** Association between demographics and co-morbidities and EDS (n = 1260).

| Variable | Unadjusted OR (95% CI) | *p*-Value (Unadjusted) | Adjusted OR (95% CI) | *p*-Value (Adjusted) |
|---|---|---|---|---|
| **Demographics** | | | | |
| Sex | | | | |
| Male | 0.81 (0.58, 1.14) | 0.2394 | 0.68 (0.37, 1.25) | 0.2154 |
| Female | 1.00 | - | 1.00 | |
| Age groups, in years | | | | |
| ≥56 years | 2.29 (1.48, 3.55) | 0.0002 | 0.50 (0.20, 1.25) | 0.1370 |
| 35–55 years | 1.02 (0.70, 1.49) | 0.9067 | 0.83 (0.45, 1.55) | 0.5675 |
| 18–35 years | 1.00 | - | 1.00 | |
| Education level | | | | |
| Less than high school | 1.21 (0.81, 1.80) | 0.3565 | - | - |
| Completed high school | 0.76 (0.47, 1.23) | 0.2664 | - | - |
| Higher than high school | 1.00 | - | - | - |
| Marital Status | | | | |
| Married/Common Law | 0.90 (0.65, 1.26) | 0.5540 | - | - |
| Single/separated/widowed/divorced | 1.00 | - | - | - |
| Body Mass Index (BMI) | | | | |
| Obese: BMI ≥ 30 | 1.21 (0.81, 1.81) | 0.3476 | 0.62 (0.36, 1.08) | 0.0907 |
| Overweight: BMI 25–29.99 | 1.18 (0.77, 1.80) | 0.4487 | 1.01 (0.60, 1.72) | 0.9573 |
| Normal: BMI < 25 | 1.00 | - | 1.00 | |
| Smoking status | | | | |
| Current smoker | 1.04 (0.58, 1.85) | 0.8963 | 0.95 (0.45, 2.03) | 0.9056 |
| Ex-smoker | 1.16 (0.57, 2.33) | 0.6869 | 1.50 (0.64, 3.50) | 0.3523 |
| Never smoker | 1.00 | - | 1.00 | |
| Alcohol use (5 or more drinks on one occasion) | | | | |
| Regularly | 0.89 (0.58, 1.36) | 0.5954 | - | - |
| Occasionally | 0.91 (0.60, 1.38) | 0.6618 | - | - |
| Never | 1.00 | - | - | - |
| Employment | | | | |
| Unemployed | 1.18 (0.82, 1.68) | 0.3720 | - | - |
| Some work | 1.00 | - | - | - |

**Table 2.** *Cont.*

| Variable | Unadjusted OR (95% CI) | p-Value (Unadjusted) | Adjusted OR (95% CI) | p-Value (Adjusted) |
|---|---|---|---|---|
| Crowding | | | | |
| >1 person per room | 1.32 (0.96, 1.81) | 0.0901 | 1.56 (1.03, 2.37) | 0.0349 |
| ≤1 person per room | 1.00 | - | 1.00 | - |
| Money left over at end of the month | | | | |
| Not enough money | 1.08 (0.73, 1.59) | 0.7070 | - | - |
| Just enough money | 0.80 (0.54, 1.19) | 0.2776 | - | - |
| Some money | 1.00 | - | - | - |
| Co-morbidities | | | | |
| Shortness of breath (SOB) | | | | |
| Yes | 2.87 (2.01, 4.09) | <0.0001 | 2.27 (1.45, 3.55) | 0.0003 |
| No | 1.00 | - | 1.00 | - |
| Loud snoring | | | | |
| Yes | 1.84 (1.25, 2.72) | 0.0020 | 2.03 (1.27, 3.25) | 0.0031 |
| No | 1.00 | - | 1.00 | - |
| Bronchitis attack | | | | |
| Yes | 1.84 (1.29, 2.60) | 0.0006 | - | - |
| No | 1.00 | - | - | - |
| Chronic lung disease | | | | |
| Yes | 2.14 (1.54, 2.96) | <0.0001 | 1.83 (1.21, 2.77) | 0.0041 |
| No | 1.00 | - | 1.00 | - |
| Sinus problem | | | | |
| Yes | 1.79 (1.27, 2.52) | 0.0008 | - | - |
| No | 1.00 | - | - | - |
| Heart Problem | | | | |
| Yes | 1.91 (1.21, 3.01) | 0.0051 | - | - |
| No | 1.00 | - | - | - |
| Tuberculosis | | | | |
| Yes | 1.65 (0.97, 2.80) | 0.0664 | - | - |
| No | 1.00 | - | - | - |
| Asthma | | | | |
| Yes | 1.16 (0.75, 1.77) | 0.5036 | - | - |
| No | 1.00 | - | - | - |
| Diabetes | | | | |
| Yes | 1.47 (0.96, 2.27) | 0.0771 | - | - |
| No | 1.00 | - | - | - |
| Depression | | | | |
| Yes | 1.99 (1.38, 2.86) | 0.0002 | 1.65 (1.05, 2.61) | 0.0314 |
| No | 1.00 | - | 1.00 | - |
| Gastric Reflux | | | | |
| Yes | 2.14 (1.49, 3.05) | <0.0001 | 2.03 (1.26, 3.25) | 0.0034 |
| No | 1.00 | - | 1.00 | - |
| Interactions | | | | |
| Sex and Age group (ref. 18–35 years) | | | | |
| Male and ≥56 years | | | 2.98 (1.33, 6.68) | 0.0080 |
| Male and 35–55 years | | | 0.93 (0.45, 1.93) | 0.8549 |
| Female and ≥56 years | | | 0.50 (0.20, 1.25) | 0.1370 |
| Female and 35–55 years | | | 0.83 (0.45, 1.55) | 0.5675 |

The mean ESS scores and prevalence of EDS at baseline (Cycle 1) and follow-up (Cycle 2) are summarized in Table 3 below. We observed that prevalence decreased over the four-year period. Based on the predicted model longitudinal changes, the prevalence of EDS was calculated for 358 individuals who participated in both cycles. The longitudinal change in the prevalence of EDS was −0.11% (that is, prevalence was decreased by 0.11%). The mean difference of predicted probability was −0.0011 ± 0.11.

**Table 3.** Mean ESS scores and prevalence of EDS at baseline (Cycle 1) and follow-up (Cycle 2).

| | Baseline (Cycle 1) Mean ESS ± SD | Prevalence of EDS Baseline (Cycle 1) | Follow-Up (Cycle 2) Mean ESS ± SD | Prevalence of EDS Follow-Up (Cycle 2) |
|---|---|---|---|---|
| Complete cases Cycle 1 & Cycle 2 | 5.45 ± 4.11 (*n* = 358) | 11.5% (41/358) | 5.09 ± 3.93 (*n* = 358) | 10.3% (37/358) |
| All cases | 5.39 ± 4.16 (*n* = 815) | 11.2% (91/815) | 5.06 ± 3.82 (*n* = 803) | 10.0% (80/803) |

## 3. Discussion

This study looked at the association between co-morbidities and the prevalence of EDS over a four-year period. The prevalence of EDS changed from 11.2% to 10.0% over a four-year period. We observed that if an individual lived in a crowded home and had any or all of the co-morbid conditions of shortness of breath (SOB), loud snoring, chronic lung disease, depression or gastric reflux, they had a higher chance of reporting EDS compared to an individual with no co-morbid conditions. This study also confirmed that individuals that were older and male were more likely to report an abnormal ESS score compared to the younger female individuals that participated. Based on the predicted model longitudinal change, the prevalence of EDS was −0.11% for 358 individuals who participated in both cycles. Stradlig et al. [34] reported that sleepiness was independently related to snoring, suggesting that snoring may reduce sleep quality sufficiently to produce substantial daytime sleepiness. Also, similar results of snoring as an independent cause of EDS were observed by Svensson et al. [35] in female populations. A study with Indigenous North Americans (Gitxsan, Nisga's and Tsimshian) living in northwestern part of British Columbia reported that frequent snoring was significantly associated with EDS [10]. Similar to these studies, we observed that individuals with loud snoring had twice the risk of EDS compared to an individual who did not experience loud snoring.

Statistics Canada has reported that 36.8% of on-reserve First Nations people live in a house with more than one person per room [36]. Another report mentioned that overcrowding contributes to social issues (lower employment rate, sleep quality, etc.) since there are many people living in one house and, as a result, individuals are forced to sleep in shifts [37]. Due to sleeping conditions, both children and adults may feel sleepy during the day as they did not have good sleep at night [37]. In this study, an individual living in a crowded house was 1.56 times more likely to report EDS compared to an individual who did not live in a crowded house.

Other studies have shown a significant association between EDS and depression [10,38–42]. An Australian study by Hayley et al. [40] reported that EDS was associated with current (Odds ratio (OR) = 2.11) and lifetime (OR = 1.95) depressive disorders. Mume et al. found that EDS was more common among depressed patients [41]. A recent Australian study of a cohort of men showed that EDS was significantly associated with depression (OR = 2.2; 95% CI: 1.3–3.8) [38]. Froese et al. [10] reported a higher prevalence of depression (44%) in a North American Indigenous population and the depression score was significantly associated with EDS. This study found a positive association between EDS and depression, which is consistent with the aforementioned studies [38–42]. Pallesen et al. postulate that depression can cause disturbances in the normal oscillations between sleep and wakefulness, which in turn can cause disturbances in the regulator pathways related to sleep and wakefulness and can lead to insomnia at night and sleepiness during the day [38]. This association between depression and EDS is explained in this study.

Similar to this study, gastroesophageal reflux has also been shown to be associated with daytime sleepiness [11,43,44]. In a study examining patients undergoing sleep studies, Guda et al. [11] found

reports that patients with symptoms of gastroesophageal reflux had a poorer sleep and consequently greater EDS compared to those without these symptoms. Results from a more recent study showed that there was a four-times greater risk of reporting daytime sleepiness with the combination of gastroesophageal reflux and snoring in women [43]. This combination resulted in EDS due to poor sleep quality related to recurrent disruption.

Certain chronic respiratory symptoms and diseases are known to be related to daytime sleepiness [17,45,46]. A recent study by Matura et al. reported that increased daytime sleepiness is associated with SOB [46], confirming the findings of others. Zohal et al. [17] reported that chronic obstructive pulmonary disease (COPD) was associated with daytime sleepiness and poor quality of sleep. Klink and Quan [18] reported that individuals with chronic bronchitis and emphysema showed a significantly higher prevalence of EDS compared to those with no airway disease. Confirming the findings of others, this study showed a significant association between EDS and chronic lung diseases.

Not many studies have looked at the prevalence of EDS in populations and related predictors over periods of time. One recent study [26] looked at predictors of incidence of EDS with a five-year follow-up and predictors of increased daytime sleepiness with a five-year follow-up; younger age, shorter sleep duration, high frequency of naps, fatigue, anxiety, sleep apnea and absence of hypnotic consumption were all found to be significant determinants of EDS at baseline. In a multivariate model, participants with increased daytime sleepiness were associated with being younger, living alone and being more depressed. In the same study, authors reported that occasional smoking, depression, fatigue and having chronic pain were predictors of incident of EDS. Another study looking at predictors for development of EDS over time reported that insomnia, anxiety and/or depression and smoking were the most important predictors of incidence of EDS [47]. This study reports that crowded homes, shortness of breath, loud snoring, chronic lung disease and depression, as well as age and sex interaction, are significant determinants of the prevalence of EDS.

*Strengths and Limitations*

One of the strengths of this present study was the prospective First Nations cohort design and large number of participants. There were a large number of potential risk factors including lifestyle, socio-demographics and medical history. In addition to that, the evaluation of EDS was assessed using the ESS, which was the most common tool used in sleep research in clinical settings [15]. This was one of the first studies that looked at longitudinal changes of EDS in two First Nation populations. However, there were limitations as well. The data were self-reported with possible recall bias, and the objective measures of sleep quality, such as the multiple sleep latency test (MSLT), were not available for this study. Another limitation was that sleep duration as a sleep habit were not collected in this study. Furthermore, the use of numerous prescription medications/drugs for co-morbid conditions can lead to EDS [3], and this study did not collect information on prescription meditation/drug use. In some situations medications used for co-morbidities can be the cause of EDS, which could be potential confounding by indication.

## 4. Materials and Methods

### 4.1. Survey Sample

The data for this study was derived from baseline assessments and follow-up evaluations (after four years) from the First Nations Lung Health Project (FNLHP), which was conducted in two Cree First Nation communities (Community A and Community B) in Saskatchewan during 2012–2013 [48] (Cycle 1) and 2016 (Cycle 2). There were 874 individuals who participated in the baseline survey (Cycle 1) and 839 who participated in the follow-up survey (Cycle 2). The follow-up rate was 45.2% (395/874). There were 859 participants aged 18 years and older at baseline (Cycle 1) and 821 participants aged 18 years and older at follow-up (Cycle 2) who completed the interviewer-administered questionnaire. Epworth Sleepiness Scale (ESS) [30–33] score availability is shown in Table 4. Both measures were

available for 358 individuals and one measurement was available for 902 individuals either in Cycle 1 or Cycle 2. The data from 1260 individuals were available for analysis.

**Table 4.** Epworth Sleepiness Scale (ESS) score patterns over two cycles *.

| Pattern | Symbol ** | N (%) |
|---|---|---|
| Complete both Cycle 1 and Cycle 2 | X  X | 358 (27.64) |
| Complete Cycle 1 only | X  - | 457 (35.29) |
| Complete Cycle 2 only | -  X | 445 (34.36) |
| None | -  - | 35 (2.70) |
| Total | | 1295 (100.00) |

* Note: Analysis is based on 358 + 457 + 445 = 1260 individuals. ESS is not available for 35 individuals. ** X: observed; -: missing.

The study was approved (on 25 April 2012) by the Biomedical Research Ethics Board of the University of Saskatchewan (Certificate No. Bio #12-189) and adhered to all of the criteria outlined in Chapter 9 entitled Research Involving the First Nations, Inuit and Metis Peoples of Canada found in the Tri-Council Policy Statement: Ethical Conduct for Research Involving Humans [49]. Written consent was obtained from all participants.

### 4.2. Data Collection

Trained community-derived research assistants carried out the baseline and follow-up interviews. Adults 18 years and older were invited to the Health Centre in each of the communities to complete the interviewer-administered questionnaires and have clinical assessments conducted. This manuscript is based on the data evolving from the interviewer-administered questionnaires. The Epworth Sleepiness Scale (ESS) [30–33] questionnaire was used to assess the degree of EDS. The ESS has not been validated in the Indigenous population [50].

### 4.3. Operational Definitions

Definition of EDS: An ESS score >10 [27,30] was considered to be abnormal and identified as a case of EDS both at baseline (Cycle 1) in 2012–2013 and at follow-up four years later (Cycle 2).

Independent variables of interest at baseline (Cycle 1) were: self-reported age; sex; body mass index (BMI) (derived from self-reported weight and height); education level; marital status; smoking status; alcohol consumption; and employment status. "Doctor ever diagnosed" conditions included: sinus trouble; heart problem; tuberculosis; attack of bronchitis; emphysema; chronic bronchitis; chronic obstructive pulmonary disease (COPD); asthma; diabetes; depression; and gastric reflux. Other factors obtained through the questionnaire included: respiratory symptoms such as chronic cough, chronic phlegm, shortness of breath (SOB); loud snoring; and money left over at end of the month. In addition, the number of persons per room as an index of crowding was included. For the analysis, the term "chronic lung disease" was used to include one or more of emphysema, chronic bronchitis, chronic cough/chronic phlegm and COPD.

### 4.4. Statistical Analysis

Statistical analyses were conducted using SAS 9.4 (SAS Institute Inc. 2018. Cary, NC, USA: SAS Institute Inc.). A Chi-squared test was used to assess relationships between abnormal ESS and covariates at baseline and p values were reported. A multilevel logistic regression model using a generalized estimating equations approach [51] was used to develop the model with individuals (first level) clustering within households (second level). Within-subject correlation between two cycles was taken into account using PROC GENMOD REPEATED SUBJECT and WITHIN SUBJECT statements. The significant contribution of potential risk factors, confounders and interactive effects was determined by developing a series of multilevel models. Variables with $p < 0.20$ in the univariate

analysis became factors for the multivariable model. The variables retained in the final multivariable model included those that were statistically significant (i.e., $p < 0.05$) as well as age, sex, BMI and smoking status. Odds ratios (ORs) and 95% confidence intervals (CIs) were used to present the strength of the associations. The QIC statistic proposed by Pan [52] was used for comparing models of the generalized estimating equations (GEE) method. When using QIC to compare two models, the model with the smaller statistic was preferred.

## 5. Conclusions

There was a decrease in prevalence of EDS over the four-year period. Some of the co-morbid conditions were associated with EDS. Therefore, managing co-morbid conditions was important to reducing the prevalence of daytime sleepiness. Further investigation into the mechanisms that influence the associations between co-morbidities and EDS will be necessary. In addition, individuals who are taking medication for different co-morbid conditions need to be studied separately to identify the true causes of the EDS.

**Author Contributions:** C.K. authored most of the paper, carried out the statistical analysis, reviewed the literature, reviewed the citations, and created the abstract and manuscript. J.D., P.P. and S.A. are the co-principal investigators of the F.N.L.H.P.R.T., C.K., D.R., S.K., S.A., M.F., J.D., and P.P. contributed to grant writing, development of study design, questionnaire development, and study coordination. V.R. provided input into the writing of the manuscript; and, edited the manuscript. J.S. served as "content experts" for the research project engaged in document review/editing and support during the data collection phases of the survey. All other co-authors significantly contributed to manuscript preparation. The First Nations Lung Health Project members contributed during the grant writing and questionnaires development and with conducting the survey. All authors read and approved the final manuscript.

**Funding:** The F.N.L.H.P.R.T. was funded by a grant from the Canadian Institutes of Health Research "Assess, Redress, Re-assess: Addressing Disparities in Respiratory Health among First Nations People", CIHR MOP-246983-ABH-CCAA-11829.

**Acknowledgments:** The First Nations Lung Health Project Team consists of: James Dosman (Designated Principal Investigator, University of Saskatchewan, Saskatoon, SK Canada); Punam Pahwa (Co-Principal Investigator, University of Saskatchewan, Saskatoon, SK Canada); Jo-Ann Episkenew (Co-Principal Investigator (deceased), Former Faculty of Indigenous Peoples' Health Research Centre, University of Regina, SK Canada), Sylvia Abonyi (Co-Principal Investigator, University of Saskatchewan, Saskatoon, SK Canada); Co-Investigators: Mark Fenton, John Gordon, Bonnie Janzen, Chandima Karunanayake, Malcolm King, Shelly Kirychuk, Niels Koehncke, Joshua Lawson, Gregory Marchildon, Lesley McBain, Donna Rennie, Vivian R. Ramsden, Ambikaipakan Senthilselvan; Collaborators: Amy Zarzeczny; Louise Hagel, Breanna Davis, John Dosman, Roland Dyck, Thomas Smith-Windsor, William Albritton; External Advisor: Janet Smylie; Project Manager: Kathleen McMullin; Community Partners: Jeremy Seeseequasis; Raina Henderson; Arnold Naytowhow; Laurie Jimmy. We are grateful for the contributions from Elders and community leaders that facilitated the engagement necessary for the study, and all participants who donated their time to participate.

**Conflicts of Interest:** The authors declare that there was no conflict of interest regarding the publication of this article. The founding sponsors had no role in the design of the study; in the collection, analyses, or interpretation of data; in the writing of the manuscript, or in the decision to publish the results.

## Abbreviations

| | |
|---|---|
| EDS | Excessive daytime sleepiness |
| ESS | Epworth Sleepiness Scale |
| FNLHP | First Nations Lung Health Project |
| COPD | Chronic obstructive pulmonary disease |
| BMI | Body mass index |
| SOB | Shortness of Breath |
| OR | Odds ratio |
| CI | Confidence interval |
| GEE | Generalized estimating equations |
| QIC | Quasi likelihood under the independence model criterion |
| MSLT | Multiple sleep latency test |

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
