# Peer review of "Association between Co-Morbidities and the Prevalence of Excessive Daytime Sleepiness over a Four-Year Period"

_2624-5175, doi:10.3390/clockssleep1040035_

Round 1

Reviewer 1 Report

Dear authors

Thank you for this intersting study. 

Few points to consider in the revision, explain in the introduction what doe comorbidity is and perhaps consider eds as a  potential risk factor. 

The methods is adequate the statistical analysis need to contol for some interaction terms, some newer epidemiological literature point to the role of this.

Strenghts and limitations need to be expanded. Comorbidities usually has medications, this was not discussed. Confounding by indication, consult epidemologist with pharmaceutical experience.

Also discuss clinical implications of this along with research recommendations.

Conclusion need to be more solid and not over stating. 

Author Response

Comments and Suggestions for Authors

Dear authors

Thank you for this interesting study. 

Few points to consider in the revision explain in the introduction what doe comorbidity is and perhaps consider eds as a potential risk factor. 

Response: Introduction is revised to incorporate the definition of co-morbidity and EDS as a potential risk factor (See page 2, para 1).

The methods is adequate the statistical analysis need to control for some interaction terms, some newer epidemiological literature point to the role of this.

Response: Possible interactions were tested (for example: smoking and BMI, sex and BMI etc.) and only significant interaction was sex and age. It was included in the final model presented in Table 2. (See also lines 217-218; page 9).

Strengths and limitations need to be expanded. Comorbidities usually has medications, this was not discussed. Confounding by indication, consult epidemiologist with pharmaceutical experience.

Response: Under the limitations, we have discussed about the prescription drugs. Please see line 146, page 7. I have revised this statement to address the concerned of the reviewer.

“Numerous prescription medications/drugs taking for co-morbid conditions can leads to EDS [3] and this study did not collect information on prescription meditation/drug use.”

Also discuss clinical implications of this along with research recommendations. Conclusion need to be more solid and not over stating. 

Response: The clinical implications of this along with research recommendations were included to the discussion (See lines 162-167, page 8). Conclusion was revised (see lines 228-229, page 10).

Reviewer 2 Report

The aim of this study was to investigate the association between co-morbidities and longitudinal changes in prevalence of EDS in adults living in two First Nations communities. However, I cannot see the data showing the longitudinal changes of EDS in this manuscript.

The aim of this study was to investigate the association between co-morbidities and longitudinal changes in prevalence of EDS in adults living in two First Nations communities. However, I cannot see the data showing the longitudinal changes of EDS in this manuscript.

This point is very critical issue in this study.

According to the object of this study, independent variable was longitudinal changes of EDS.

How many subjects with EDS at follow-up (cycle 2) who had no EDS at baseline survey (cycle 1)? There were 358 individuals who had ESS data both survey (cycle 1 and 2) so data sample should be 358 in order to investigate the longitudinal changes of EDS.

Table 2 showed the association between demographic and co-morbidities and EDS in 1260 sample. Where is the result of longitudinal change of EDS? As the authors mentioned the participants who had ESS were 457 (cycle 1 only) and 445 (cycle 2 only), how can I know the change of EDS on them without ESS data? The sample of this study was just gathering of two different times (cycle 1 and 2) and the authors analyzed them in order to determine the factors of the EDS.

Sleep duration as sleep habit is at very least to have data in this kind of study, so this is another main limitation in case of no data set. 

I recommend to analyze again the 358 individuals in order to perform the objective of this study and show the change of EDS like as ref no. 23 (Incidence, worsening and risk factors of daytime 302 sleepiness in a population-based 5-year longitudinal study. Sci Rep 2017).

Author Response

Comments and Suggestions for Authors

 The aim of this study was to investigate the association between co-morbidities and longitudinal changes in prevalence of EDS in adults living in two First Nations communities. However, I cannot see the data showing the longitudinal changes of EDS in this manuscript. This point is very critical issue in this study.

According to the object of this study, independent variable was longitudinal changes of EDS.

Response:

I have changed the title and objectives of the paper to reflect the analysis carried out here to “Association between co-morbidities and the prevalence of excessive daytime sleepiness over four-year period”.

How many subjects with EDS at follow-up (cycle 2) who had no EDS at baseline survey (cycle 1)?

Response:

This is information is presented in “Chandima P. Karunanayake, James A. Dosman, Sylvia Abonyi, Joshua Lawson, Donna Rennie, Shelley Kirychuk, Mark Fenton, Vivian Ramsden, Jeremy Seeseequasis, Punam Pahwa and The First Nations Lung Health Project Research Team. Incidence of daytime sleepiness and associated factors in two First Nations communities in Saskatchewan, Canada. Clocks & Sleep 2018; 1, 13–25. doi:10.3390/clockssleep1010003” before, but I have included this statement with the reference (see page 2, lines 62-64).

 “Of the 358 participants who have both ESS measurements, 317 who did not have EDS at baseline and 41 did have EDS at baseline. There were 24 subjects with EDS at follow-up (cycle 2) who had no EDS at baseline survey (cycle 1)[29].The incidence was 24/317=7.6% [29].”

There were 358 individuals who had ESS data both survey (cycle 1 and 2) so data sample should be 358 in order to investigate the longitudinal changes of EDS.

Table 2 showed the association between demographic and co-morbidities and EDS in 1260 sample. Where is the result of longitudinal change of EDS? As the authors mentioned the participants who had ESS were 457 (cycle 1 only) and 445 (cycle 2 only), how can I know the change of EDS on them without ESS data? The sample of this study was just gathering of two different times (cycle 1 and 2) and the authors analyzed them in order to determine the factors of the EDS.

I recommend to analyze again the 358 individuals in order to perform the objective of this study and show the change of EDS like as ref no. 23 (Incidence, worsening and risk factors of daytime 302 sleepiness in a population-based 5-year longitudinal study. Sci Rep 2017).

Response:

I agreed with reviewer that to see the longitudinal change of EDS, we need to consider 358 participants with both measurements. The prediction Model was based on 1260 individuals in order to determine factors of the EDS. Based on the predicted model longitudinal changes in the prevalence of EDS was calculated on 358 individuals who participated in both cycles. The longitudinal change in the prevalence of EDS is -0.11%; that is prevalence was decreased by 0.11%. [The mean difference of predicted probability is -0.0011 +/- 0.11]. This additional analysis was presented in the paper. In addition to that Table of mean ESS scores and the Prevalence of EDS at baseline (Cycle 1) and follow-up (Cycle 2) is included.

Baseline (Cycle 1)

Mean ESS +/- SD

Prevalence of EDS

Baseline (Cycle 1)

Follow-up (Cycle 2)

Mean ESS +/- SD

Prevalence of EDS

Follow-up (Cycle 2)

Complete cases Cycle 1 & Cycle 2

5.45 +/- 4.11

(n=358)

11.5%

(41/358)

5.09 +/- 3.93

(n=358)

10.3%

(37/358)

All Cases

5.39 +/- 4.16

(n=815)

11.2 %

(91/815)

5.06 +/- 3.82

(n=803)

10.0%

(80/803)

As mentioned in the previous comment, the incidence of EDS is previously presented in “Chandima P. Karunanayake, James A. Dosman, Sylvia Abonyi, Joshua Lawson, Donna Rennie, Shelley Kirychuk, Mark Fenton, Vivian Ramsden, Jeremy Seeseequasis, Punam Pahwa and The First Nations Lung Health Project Research Team. Incidence of daytime sleepiness and associated factors in two First Nations communities in Saskatchewan, Canada. Clocks & Sleep 2018; 1, 13–25. doi:10.3390/clockssleep1010003”. Therefore, I have included a statement with the reference.

Sleep duration as sleep habit is at very least to have data in this kind of study, so this is another main limitation in case of no data set. 

Response:

Thanks for the comment. I have included unavailability of sleep duration data as another major limitation.

Round 2

Reviewer 2 Report

The title and objective of this study were revised. I think this change is appropriate.

There is one minor issue.

Please check the first sentence out. Was prevalence decreased or increased over four year period?

“We observed that prevalence was increased over four-year period. Based on the predicted model longitudinal changes in the prevalence of EDS was calculated on 358 individuals who participated in both cycles. The longitudinal change in the prevalence of EDS was -0.11%; that is prevalence was decreased by 0.11%.”

Author Response

Comments and Suggestions for Authors

The title and objective of this study were revised. I think this change is appropriate.

Response: Thank you.

There is one minor issue.

Please check the first sentence out. Was prevalence decreased or increased over four year period?

“We observed that prevalence was increased over four-year period. Based on the predicted model longitudinal changes in the prevalence of EDS was calculated on 358 individuals who participated in both cycles. The longitudinal change in the prevalence of EDS was -0.11%; that is prevalence was decreased by 0.11%.”

Response: Thanks for noticing the mistake. The prevalence was decreased over four year period. Sentence was corrected as follows:

“We observed that prevalence was decreased over four-year period.”